# Development of a mouse model of ascending infection and preterm birth

**Nicholas R. Spencer**[1][☉], **Enkhtuya Radnaa**[1][☉], **Tuvshintugs Baljinnyam**[2],
**Talar Kechichian**[1], **Ourlad Alzeus G. Tantengco**[1,3], **Elizabeth Bonney**[4], **Ananth
Kumar Kammala**[1], **Samantha Sheller-Miller**[1], **Ramkumar Menon**[1]*

1 Division of Maternal-Fetal Medicine and Perinatal Research, Department of Obstetrics and Gynecology,
The University of Texas Medical Branch at Galveston, Galveston, Texas, United States of America,
2 Department of Pharmacology and Toxicology, The University of Texas Medical Branch at Galveston,
Galveston, Texas, United States of America, 3 Department of Biochemistry and Molecular Biology, College of
Medicine, University of the Philippines Manila, Manila, Philippines, 4 Department of Obstetrics and
Gynecology, University of Vermont, Burlington, VT, United States of America

☉ These authors contributed equally to this work.
* ra2menon@utmb.edu

pone.0260370

CANADA

**Data Availability Statement:** All relevant data are
within the paper and its Supporting Information
files.

## Abstract

### Background

Microbial invasion of the intraamniotic cavity and intraamniotic inflammation are factors
associated with spontaneous preterm birth. Understanding the route and kinetics of infec-
tion, sites of colonization, and mechanisms of host inflammatory response is critical to
reducing preterm birth risk.

### Objectives

This study developed an animal model of ascending infection and preterm birth with live bac-
teria (*E. coli*) in pregnant CD-1 mice with the goal of better understanding the process of
microbial invasion of the intraamniotic cavity and intraamniotic inflammation.

### Study design

Multiple experiments were conducted in this study. To determine the dose of *E. coli* required
to induce preterm birth, CD-1 mice were injected vaginally with four different doses of *E. coli*
($10^3$, $10^6$, $10^{10}$, or $10^{11}$ colony forming units [CFU]) in 40 µL of nutrient broth or broth alone
(control) on an embryonic day (E)15. Preterm birth (defined as delivery before E18.5) was
monitored using live video. *E. coli* ascent kinetics were measured by staining the *E. coli* with
lipophilic tracer DiD for visualization through intact tissue with an in vivo imaging system
(IVIS) after inoculation. The *E. coli* were also directly visualized in reproductive tissues by
staining the bacteria with carboxyfluorescein succinimidyl ester (CFSE) prior to administra-
tion and via immunohistochemistry (IHC) by staining tissues with anti-*E. coli* antibody. Each
pup's amniotic fluid was cultured separately to determine the extent of microbial invasion of
the intraamniotic cavity at different time points. Intraamniotic inflammation resulting from *E.
coli* invasion was assessed with IHC for inflammatory markers (TLR-4, P-NF-κB) and neu-
trophil marker (Ly-6G) for chorioamnionitis at 6- and 24-h post-inoculation.

**Funding:** R.M. Grant Support from ILIAS Biologics & ILIAS Therapeutics, Inc., USA The funders had no role in study design, data collection and analysis, decision to publish, or preparation of the manuscript.

**Competing interests:** The authors have declared that no competing interests exist.

## Results

Vaginally administered *E. coli* resulted in preterm birth in a dose-dependent manner with higher doses causing earlier births. In *ex vivo* imaging and IHC detected uterine horns proximal to the cervix had increased *E. coli* compared to the distal uterine horns. *E. coli* were detected in the uterus, fetal membranes (FM), and placenta in a time-dependent manner with 6 hr having increased intensity of *E. coli* positive signals in pups near the cervix and in all pups at 24 hr. Similarly, *E. coli* grew from the cultures of amniotic fluid collected nearest to the cervix, but not from the more distal samples at 6 hr post-inoculation. At 24 hr, all amniotic fluid cultures regardless of distance from the cervix, were positive for *E. coli*. TLR-4 and P-NF-κB signals were more intense in the tissues where *E. coli* was present (placenta, FM and uterus), displaying a similar trend toward increased signal in proximal gestational sacs compared to distal at 6 hr. Ly-6G+ cells, used to confirm chorioamnionitis, were increased at 24 hr compared to 6 hr post-inoculation and control.

## Conclusion

We report the development of mouse model of ascending infection and the associated inflammation of preterm birth. Clinically, these models can help to understand mechanisms of infection associated preterm birth, determine targets for intervention, or identify potential biomarkers that can predict a high-risk pregnancy status early in pregnancy.

## Introduction

Spontaneous preterm birth (PTB) and preterm prelabor rupture of the fetal membranes (pPROM) are major complications of pregnancy that impact ~ 11% of all pregnancies around the globe [1]. PTB and pPROM are associated with several risk factors, including genetic, race/ethnicity, geographic location, socio-economic status, prior history, and family history, all of which may impact more than one pregnancy over time [2, 3]. Conversely, risk factors such as maternal and intraamniotic infections, inflammation, behavioral, vascular, and endocrine dysfunctions during pregnancy can generate very complex dynamic biochemical and/or biomechanical pathways that can manifest as PTB or pPROM [2, 4] during an index pregnancy. Maternal and fetal infections and host inflammatory responses are associated with ~ 50% of all PTB and 70% of all pPROM [2, 3]. Infection and host inflammatory responses are also contributors to various morbidities in preterm neonates [5–8]. Cerebral palsy, periventricular leukomalacia, enterocolitis, and the autism spectrum of disease are linked to infection and infection-associated host inflammatory responses [9–15]. Understanding these dynamic risk factors and their interactions with feto-maternal uterine tissues is critical to reducing the incidence of PTB and pPROM and associated morbidities.

Isolation of microbes from various feto-maternal tissues and increased presence of inflammatory mediators in amniotic fluid, cord and maternal plasma, and cervico-vaginal fluid of women with PTB or pPROM indicates a mechanism by which disease manifests via ascending infection [16–23]. Ascending vaginal infection leading to microbial invasion of the intraamniotic cavity (MIAC) and the establishment of intraamniotic infection and inflammation (IAI) is the most hypothesized path of infection associated PTB and pPROM [3, 24–26]. Systemic maternal infections (e.g., periodontal disease, urinary infections, bacteremia) [27–29] or the

introduction of microbes directly to the amniotic cavity during invasive procedures (e.g., fetal surgery, chorionic villous sampling, amniocentesis) are also ways by which microbial colonization initiates [15, 24, 30–32]. Antimicrobial interventions have not been successful in reducing the incidence of PTB or pPROM and are often controversial due to developmental impacts on children exposed to antibiotics in utero [33–35]. These outcomes suggest gaps in our current understanding of how pathogen propagation and colonization at sterile sites mechanistically induce labor or cause membrane rupture.

The severity of infection and host inflammatory response are dependent on the type of pathogen, their load, and polymicrobial etiology [36–38]. Microbial isolates and proinflammatory markers (cytokines, chemokines, matrix degrading enzymes etc.) are often similar in both PTB with intact membranes and pPROM [39]. In cases with documented intraamniotic infection, the dichotomy between some women delivering preterm with intact membranes and others with pPROM suggests that intrauterine colonization and/or infection (mechanisms, functional pathways, and biomarkers) can produce distinct pathologic pathways and outcomes in different subjects [40–45]. In pPROM, microbial colonization has been associated with membrane weakening due to collagen rich extracellular matrix degradation [46]. However, it has been reported that bacterial collagenases are not specifically designed to degrade human collagens. This may suggest that endogenous activation of host inflammatory response is essential to cause the pathologic changes observed [47, 48]. The precise pathologic mechanisms that can lead to infection-associated diverse pregnancy complications are still unclear. A reliable model can advance our knowledge and help to develop strategies to mitigate the risk of infection and inflammation associated with PTB and pPROM.

Animal models have been reliably used to address questions related to infection during pregnancy. Intraperitoneal or intrauterine lipopolysaccharide (LPS) and other microbial antigen injection has been used traditionally to mimic infection by many laboratories including our own [49–55]. Although several valuable pieces of information have been generated, the doses and route of LPS administration employed by these studies may bias innate immune responses and bypass natural host defense mechanisms. Therefore, the effects of these experimentally produced exposures may not completely mimic the conditions associated with human infection-associated PTB. To overcome this limitation, several animal models have been created either with live ascending infection or systemic infection to understand relevant mechanisms. A recent classic report by Suff *et al.* demonstrated that intravaginal administration of two bioluminescent strains *E. coli*, a non-pathogenic and another pathogenic, induced preterm delivery and development of fetal neuroinflammation in response to an ascending infection model [56].

Reliable and reproducible models are needed to show MIAC and IAI. In this study, we recreated an ascending infection model with vaginal inoculation of *Escherichia coli (E. coli)* in a CD-1 mouse model of pregnancy. We determined the bacterial dose-dependent pregnancy outcome (PTB), the kinetics of ascension, route of transmission, and development of uterine tissue inflammation. Clinically, these models are expected to improve the quality of studies that will determine various agents of intervention to reduce the risk of adverse pregnancy outcomes, avoid interventions that may not be beneficial during pregnancy, and generate potential biomarkers to predict high risk pregnancy status.

## Materials and methods

### Mouse model of preterm birth

All animal procedures were approved by the Institutional Animal Care and Use Committee (IACUC) at the UTMB. Timed pregnant CD-1 mice were purchased from Charles River Laboratories (Houston, TX, USA) and received on a gestational day 14 (E14) and were housed in a

temperature and humidity-controlled facility with 12:12-h light and dark cycles. On E15, pregnant mice were anesthetized deeply with inhalation of isoflurane and subjected to vaginal administration of bacteria by delivering 40 μL of bacterial suspension using 200-ml pipet tips. A volume of 40 μL was chosen based on previous studies showing vaginal administration without leakage while containing the selected CFU. As controls, the same volume of sterile nutrient broth (Difco™ nutrient broth, BD Biosciences, Cat. # BD234000, Lot. 9219600) was administered for all experiments. Animals were continuously monitored using Wansview cameras (Shenzhen Wansview Technology Co., Ltd, Shenzhen, China) to determine the timing of delivery.

## Escherichia coli (*E. coli*) culture

The strain of bacteria used in this study is ATCC 12014 Escherichia coli O55:K59(B5):H- obtained from Remel Laboratory of Thermo-Fisher (Thermo Fisher Scientific, Remel Products, Lenexa, KS, USA, Lot# 496291). The bacteria were cultured in sterile, non-selective nutrient broth (BD Biosciences) and stocks were stored at -80˚C in 20% glycerol.

**Dose determination for *E. coli* inoculation.** To introduce bacteria at the specified doses, we generated a standard curve to predict live bacteria quantity based on their colony forming unit (CFU) [57, 58]. For each independent experiment, 0.5 ml of bacterial stock was transferred to 200 ml of Luria Broth (LB) and cultured for 16 hr at 37˚C with 200 rpm agitation. On the day of experiment, $OD_{600}$ value was determined for the culture in triplicate measurements with a spectrophotometer (D30 BioPhotometer, Eppendorf, Hamburg, Germany). Using the average value of $OD_{600}$, we estimated the CFU for that culture via a predetermined formula [required volume (mL) = (target (CFU) x number of animals) / current (CFU/mL)]. Occasionally, we prepared 1.5x of the needed volume for the target CFU by transferring the bacterial culture to a centrifuge tube and centrifuging at 4000 xg for 10 min. This pellet was resuspended in a pre-determined volume of LB. In these circumstances, the main fraction of culture (1x) was used for the animal experiment and remaining fraction (0.5x) was diluted and spread on LB-agar plates (Difco™ nutrient agar, BD Biosciences, San Jose, CA, USA, Cat. # BD213000, Lot. 9218604) (in triplicates). After overnight culture at 37˚C, the colonies were counted and the CFU were calculated. Finally, actual CFU was compared to the target CFU and throughout the study the variability of the CFUs were within ±18% from the target CFU [58].

**Preparation of *E. coli* for vaginal administration.** On the day before the experiment, 0.5 mL of frozen bacterial stock was transferred to 200 mL of nutrient broth and cultured overnight at 37˚C with shaking at 200 rpm. Sixteen h later, an $OD_{600}$ was measured in order to calculate the equivalent CFU, then the required volume of culture was transferred to a centrifuge tube and spun at 4000g for 10 min. After centrifugation, the aqueous phase was carefully removed without disturbing the pellet, and the resulting bacterial pellet was resuspended in a predetermined volume of sterile broth and introduced to the mouse, as described above.

## Determination of dose required to induce preterm birth

Bacteria were administered vaginally to the mice on E15 at varying doses ($10^3$, $10^6$, $10^{10}$, and $10^{11}$ CFU) or equivalent volume of sterile nutrient broth as control. Mice were then recorded via camera and timing of delivery (defined as delivery of first pup) was documented. Delivery on or before E18.5 contributing to developmentally immature pups was considered PTB [59].

## DiD staining of *E. coli* for IVIS imaging

Lipophilic tracer DiD (DiIC18(5); 1,1′-dioctadecyl-3,3,3′,3′- tetramethylindodicarbocyanine, 4-chlorobenzenesulfonate salt; Invitrogen, Thermo Fisher Scientific, Carlsbad, CA, Cat. # D7757, Lot. 2186103) was used to stain the bacteria for subsequent imaging to monitor

ascending infection through the uterine cavity. We used DiD to label *E. coli* for whole tissue imaging with IVIS to avoid autofluorescence and phototoxic effects. Bacteria ($10^{11}$ colony forming units, CFU) were stained with either 100 μM or 500 μM DiD for 30 min at 37˚C, and excess dye was washed three times with broth. The bacteria were centrifuged at 4000 g for 10 minutes for each wash. DiD-stained bacterial pellets were resuspended in 40 μL of broth and administered vaginally to E15 mice as described above. Mice were sacrificed at 6 hr after the bacterial administration because our ascending infection kinetic showed bacterial invasion within 6 hr, and the reproductive organs, whole intact uterus, including the embryos, were removed, and transferred on ice to Biomedical Imaging Facility, UTMB, and imaged with IVIS Spectrum CT In Vivo Imaging System (PerkinElmer, Waltham, MA, USA). After imaging the uterus, proximal and distal embryos were removed from the uterine cavity and the placenta, fetal membrane, embryo, and cervix were imaged separately with IVIS.

## Carboxyfluorescein succinimidyl ester (CFSE) staining of *E. coli* for immunofluorescence imaging

For *E. coli* detection in the tissue sections with histology, we used fluorescent dye CFSE to label the bacteria. Bacteria ($10^{10}$ CFU) were stained with 10 μM of CFSE (Invitrogen, Carlsbad, CA, USA, Cat. # 65-0850-84, Lot. 2178212) for 30 min at RT, washed three times with broth, and spun down at 4000 xg for 10 min for each wash. The bacterial pellets were resuspended in 40 μL of broth and vaginally administered to E15 mice as described above. Mice were then sacrificed at 6, 24, or 48 hr after bacterial administration. Frozen sections of reproductive tissues (cervix, uterus, placenta, fetal membrane, and pups) were subjected to microscopic analysis for the CFSE signal as described previously [60]. Briefly, the collected tissues were fixed in 4% PFA overnight at 4˚C, then incubated in 30% sucrose for additional 24 hr at 4˚C for cryoprotection. The next day, tissues were embedded in Tissue-Tek optimal cutting temperature (OCT) compound (Sakura Finetek, Tokyo, Japan). Sections (10 μM) were air-dried at RT for 40 min to allow tissues to adhere to precoated hydrophilic slide glasses (Matsunami Glass, Osaka, Japan). After washing with TBS-T (Tris-buffered Saline + Tween-20) to remove OCT, the sections were stained with DAPI for 5 min, mounted with Mowiol (Calbiochem, San Diego, CA, USA, Cat. # 475904), and visualized with a Keyence microscope (Keyence Corp., Osaka, Japan). Images were analyzed with BZ-X800 Analyzer (Keyence Corp).

## Bacterial culture from amniotic fluid and maternal blood

A dose of $10^{11}$ CFU *E. coli* versus sterile nutrient broth was administered vaginally to E15 mice. The mice were sacrificed 6 or 24 hr after administration. Maternal blood was collected from the right ventricle, cooled on ice for 30 minutes, and centrifuged at 2000g for 10 min. The serum was collected and stored at -80˚C and a bacterial loop full of blood was cultured on MacConkey agar (Millipore Sigma, Louis, MO, USA, Cat. # M7408-250G), incubated at 37˚C, and examined and photographed on days 1 and 5 for microbial growth. The amniotic fluid from each gestational sac was collected using aseptic technique and centrifuged at 4000g for 10 minutes. The supernatant was collected and stored at -80˚C and the pellet was resuspended in sterile Endotoxin-Free Dulbecco's PBS (1X) (w/o Ca++ & Mg++) (Millipore Sigma, Cat. # TMS-012-A), cultured on MacConkey agar, incubated at 37˚C, and examined and photographed on days 1 and 5.

## Tissue collection for formalin-fixed paraffin embedded (FFPE)

Each pup, fetal membranes, placenta, and corresponding uterine segment were separately stored in 10% neutral buffered formalin for 30–48 hr. The tissues were then processed through

gradient ethanol for dehydration followed by xylene and were embedded in paraffin as reported previously [61–63].

## Immunohistochemistry (IHC)

Paraffin-embedded sections were cut 5 µm thick, mounted on precoated hydrophilic glass slides (Matsunami Glass), dried at 37˚C to ensure adherence to the slides, and stored at 4˚C until use. Sections were baked at 50˚C overnight before staining. Paraffin sections were deparaffinized in 3 changes of xylene for 10 min each, then rehydrated through a series of graded alcohols with a final rinse in distilled water. Sections were then subjected to antigen retrieval by heating at 121˚C in citrate buffer for 20 min. The slides were then rinsed in distilled water, TBS, and blocked with 3% BSA/TBS-T for 1 h at RT. Then, tissues were stained with anti—*E. coli* antibody (Abcam, Cambridge, MA, USA, Cat. # ab137967, 1:1000 df), anti-Ly-6G/Ly-6C antibody (RB6-8C5) (Novus Biologicals, Littleton, CO, USA, Cat. # NBP2-00441, 1:100 df) and anti-TLR-4 antibody (Novus Biologicals, Littleton, CO, USA, Cat. # NBP2-24821, 1:200 df) diluted in 3% BSA/TBS-T overnight at 4˚C. The next day, the tissues were washed then incubated with secondary antibody (Alexa Fluor® 594, Abcam, Cat# ab150080, Lot: GR3323881-1 with 1:1000 df for *E. coli*, TLR-4, and DyLight 650, Novus Biologicals, Cat # NBP2-60688C, Lot: 39933-102120-c with 1:1000 df for Ly-6G/Ly-6C) for 2 h at RT followed by DAPI staining. Images were obtained and analyzed as described above using the BZ-X800 Analyzer (Keyence Corp).

## Gentamicin administration

To determine if antibiotic intervention can delay ascending infection-induced PTB, $10^{11}$ CFU *E. coli* infected mice were treated with a single dose of 20mg/kg of gentamicin [64] via tail vein intravenous injections 4 and 24 hr after the exposure to the bacteria. Mice then were video monitored until delivery.

## Statistical analysis

Statistical analysis was performed using the GraphPad Prism 8.0 software (GraphPad, San Diego, CA). Statistical parameters associated with the figures are reported in the figure legends. All data are reported as the mean ± SEM. Statistical significance in differences between experimental groups to controls was assessed as following: unpaired t-test for *E. coli* dose dependent preterm birth study, paired t-test for neutrophils quantification and Fisher's exact test for the rates of preterm birth. Significance was considered at $P < 0.05$.

# Results

## Dose dependent induction of preterm birth by *E. coli*

As shown in **Fig 1**, animals injected with liquid broth (LB, plain microbial culture medium—control) delivered at term whereas a dose dependent shortening of time interval to delivery was seen in *E. coli* injected animals. Time to delivery after administration of varying doses of *E. coli* demonstrated shorter latency periods with increasing doses of *E. coli*. Administration of $10^3$ and $10^6$ CFU *E. coli* induced PTB in 78.4 ± 8.9 and 64 ± 18.7 h, respectively whereas $10^{10}$ CFU induced PTB within 48 hr (41.73 ± 7.5) and $10^{11}$ CFU induced preterm birth within 30 h (29.5 ± 6.3) of *E. coli* administration (**S1 Fig**). PTB produced non-viable pups regardless of dose. Phenotypic outcome (PTB) was our primary objective and fetal tissue inflammation, or other maternal or neonatal clinical outcomes were not determined.

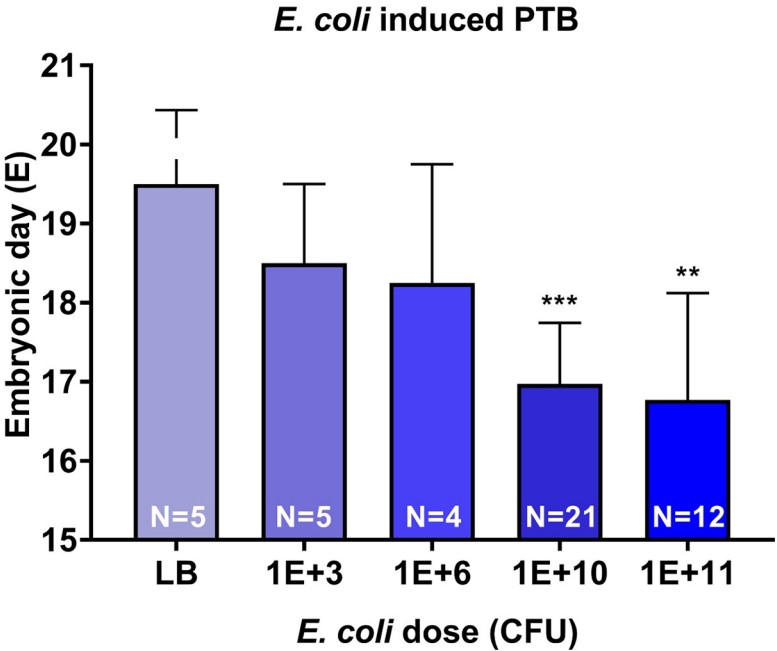

**Fig 1. *E. coli* induced preterm birth (PTB) in dose dependent manner.** Higher dose of *E. coli* ($10^{11}$ CFU and $10^{10}$ CFU) significantly shortened gestational day of delivery compared to control (LB) ($P = 0.0012$ and $P<0.001$, respectively). Low dose of *E. coli* ($10^6$ CFU and $10^3$ CFU) shortened gestational day of delivery compared to control, however it was not significant ($P = 0.17$ and $P = 0.2$, respectively). CFU- colony forming unit.

## Vaginally administered *E. coli* ascends through the uterine cavity

IVIS imaging detected no signal from reproductive tissues after administration of sterile liquid broth (microbial culture media -control) or unstained *E. coli*. DiD dye alone showed limited signal at the cervix; however, DiD stained *E. coli* showed stronger signal intensity in the cervix and proximal portions of the uterine cavity, but not in the distal portion (**Fig 2**). When the reproductive tissues were imaged separately, strong DiD signal was observed in the placenta, fetal membranes, and fetus of the most proximal to cervix, but not in the same tissues of the most distal (to cervix) gestational tissues. This suggested that the bacteria carrying the dye invaded from the vagina and through the proximal reproductive tissues before reaching the distal portion of the uterus.

When tissues from animals injected with CFSE-stained *E. coli* were examined, CFSE signal was detected in the uterus, fetal membranes, and placenta of the infected mice compared to control. This signal was detectable at 6 hr after administration but became stronger at 24 hr. The signal was noted to become diffuse, likely through death and division of stained *E. coli* (**S2 Fig**). To better localize the *E. coli* within gestational tissues, we used an anti-*E. coli antibody and* immunohistochemistry. This method showed the presence of an *E. coli* signal in the cervix, uterus, fetal membranes, and pups (**Figs 3A, 4B, 5A, 5D and S3 Fig**).

At 6 hr after inoculation, the uterus and fetal membranes showed signal in proximal gestational tissues, but not in the distal tissues (**Fig 5A and 5D**). Interestingly, the oropharynx of the pup nearest to cervix showed colonization by *E. coli*, but the pup farthest from the cervix showed no signal (**S3 Fig**).

## Recovery of *E. coli* from amniotic fluid

Amniotic fluid samples collected from each horn on both sides were cultured on MacConkey's agar for testing for microbial growth. Colonies of *E. coli* showing classic characteristics of *E.*

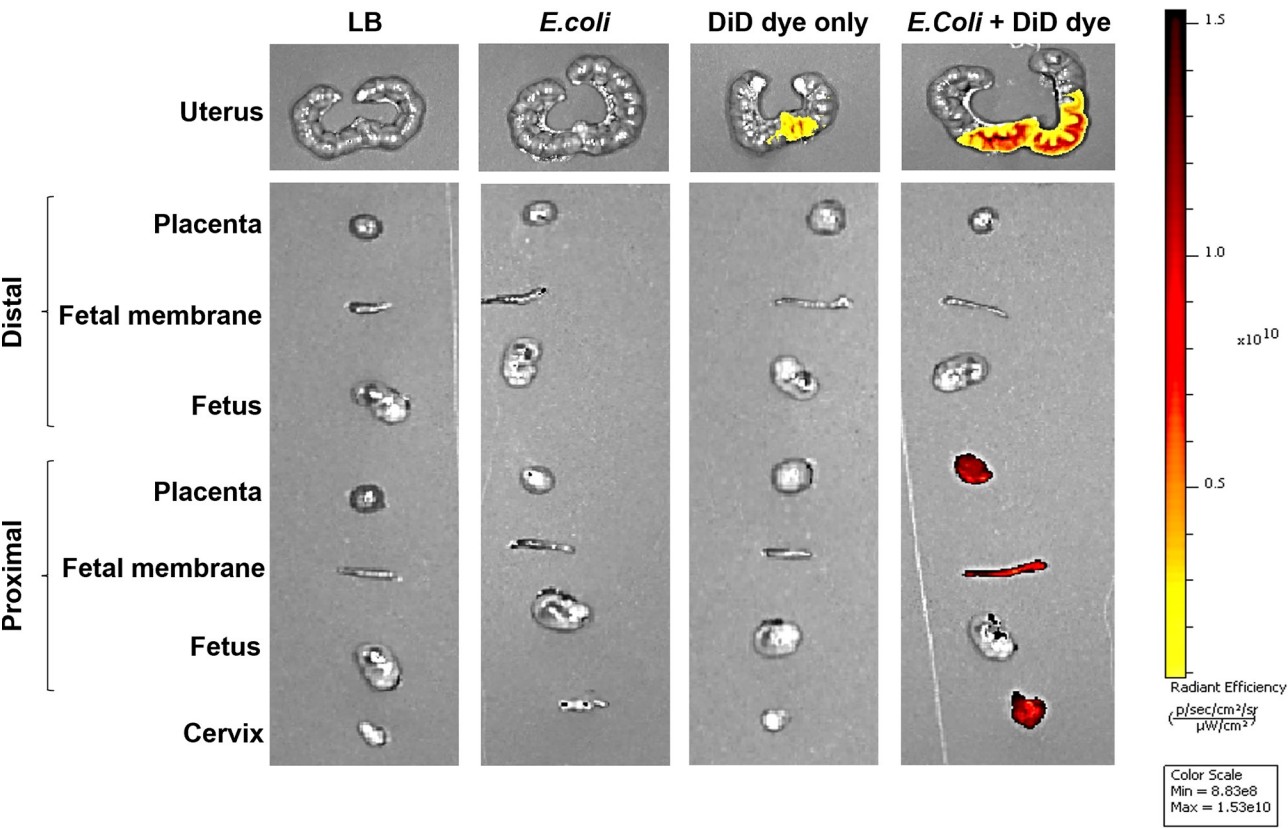

**Fig 2. DiD-stained *E. coli* ascending infection in the uterine cavity with ex-vivo IVIS imaging.** Fetuses, fetal membranes, and placentas collected from proximal to cervix and distal to cervix were imaged with in vitro imaging system (IVIS) 6 hr after vaginal administration of $10^{11}$ CFU of *E. coli* to E15 pregnant mice (N = 3). Cervix and proximal placentas, fetal membranes and fetuses show bacterial invasion (red), while distal organs show no bacterial invasion. DiD—(DiIC18(5); 1,1′-dioctadecyl-3,3,3′,3′- tetramethylindodicarbocyanine, 4-chlorobenzenesulfonate salt).

*coli* were seen (dry, flat, round, and lactose fermenting). None of the amniotic fluid from control mice grew colonies on the MacConkey agar (**Fig 3B**). Additionally, none of the maternal blood cultures showed growth, indicating absence of maternal bacteremia (**S5A and S5B Fig**). However, when the amniotic fluid from each gestational sac was cultured separately, it was noted that the proximal amniotic fluid specimens formed colonies more often than the distal amniotic fluid specimens (**Fig 3B**, **S4C Fig**). The farthest pups (from cervix) were negative in the first 6 hr compared to proximal ones. However, cultures were positive in all amniotic samples collected at 24 hr, suggesting microbial invasion in all amniotic cavities (**Fig 3B**). This suggests that bacteria invade the amniotic sacs in a stepwise sequential fashion from proximal to distal intraamniotic cavities (**Fig 4A**).

## Vaginally administered *E. coli* induces inflammation

Serial sections and immunohistochemistry were used to localize *E. coli*, as well as to show the presence of inflammatory markers in the same region. As shown in **Fig 4B**, *E. coli* was seen in cervical tissues within 6 hr and TLR-4 and P-NF-κB was also localized in the same region (**Fig 4C and 4D**). Positively stained inflammatory markers were detected in sections from *E. coli*-injected animals than in sections from control media-injected animals. Similar localization using immunohistochemical staining was done in uterine and fetal membrane tissues (**Fig 5**). As shown in **Fig 5A**, the number of cells positive for *E. coli* was much higher in the proximal

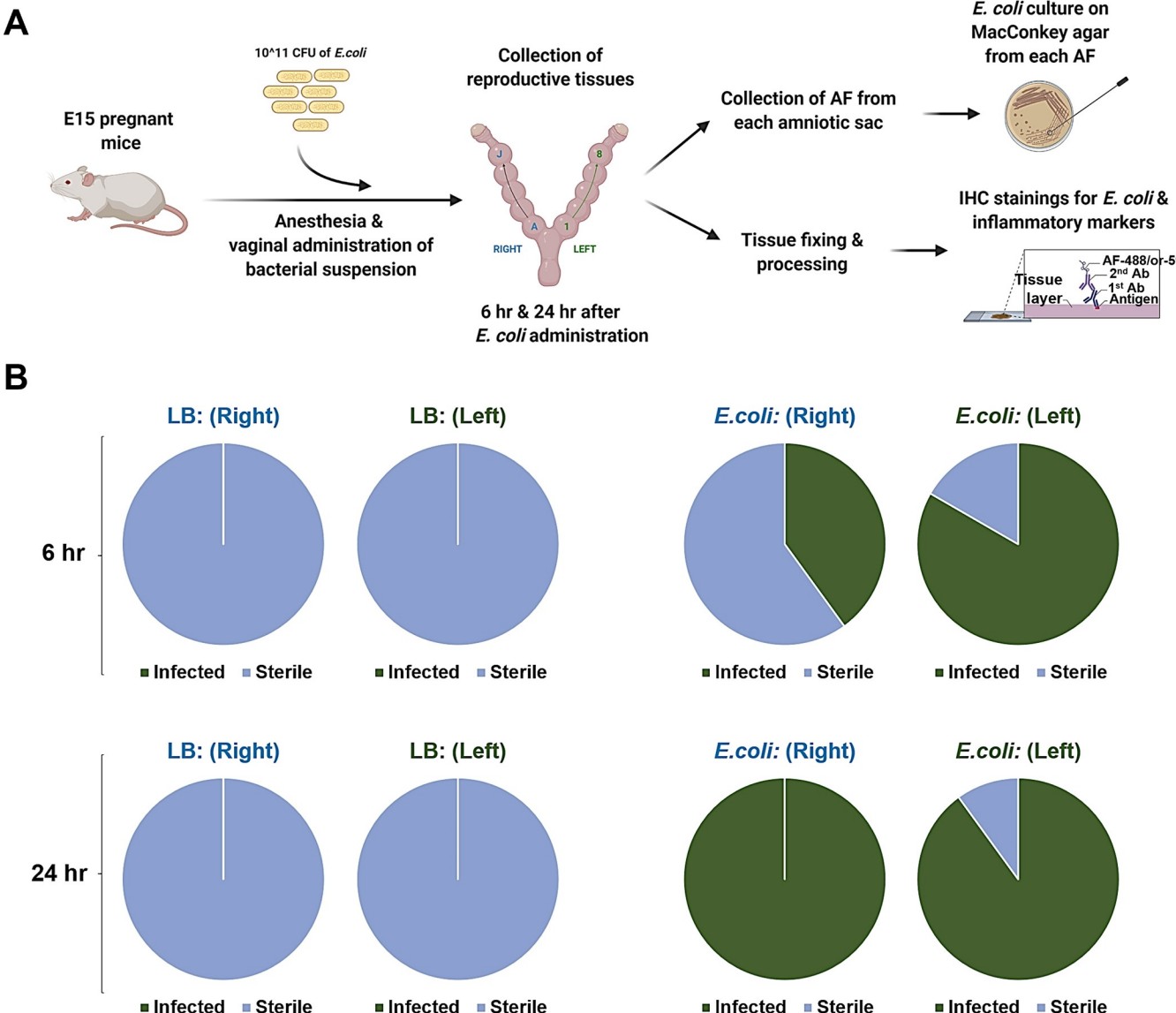

**Fig 3. *E. coli* ascending invasion occurs in a stepwise fashion from proximal to distal embryos. (A)** Graphic illustration of ascending infection evaluation via MacConkey agar culture and immunohistology. CFU- colony forming unit. AF- amniotic fluid. Reprinted from biorender under a CC BY license, with permission from biorender, original copyright. **(B)** *E. coli* culture on MacConkey agar using amniotic fluid collected from each separate embryo. Control culture shows no growth of bacteria (blue) for both 6 and 24 hr after bacterial administration. Culture of amniotic fluid from *E. coli*-administered (10¹¹ CFU) mice indicates incomplete invasion at 6 hr (half-way green) for both uterine horns (L-left, R-right) and complete invasion (full-way green) at 24 hr for both uterine horns (L and R).

uterus within 6 hr. Expression of both TLR-4 and P-NF-κB were also higher in proximal compared to distal horns (**Fig 5B and 5C**). Liquid broth (control) injected animal tissues remained negative for *E. coli* as expected with minimal levels of TLR-4 and P-NF-κB. This is expected as a low-level inflammation is expected in these tissues on E15 and E16 as the process of labor is expected to begin around this time in this model. **Fig 5D and 5F** shows a similar trend in fetal membranes. *E. coli* reached the fetal membranes of proximal horns within 6 hr (**Fig 5D**) along with increased TLR-4 (**Fig 5E**) and P-NF-κB (**Fig 5F**). As seen in the uterus, distal horns showed a weak staining suggesting time dependency of microbial invasion.

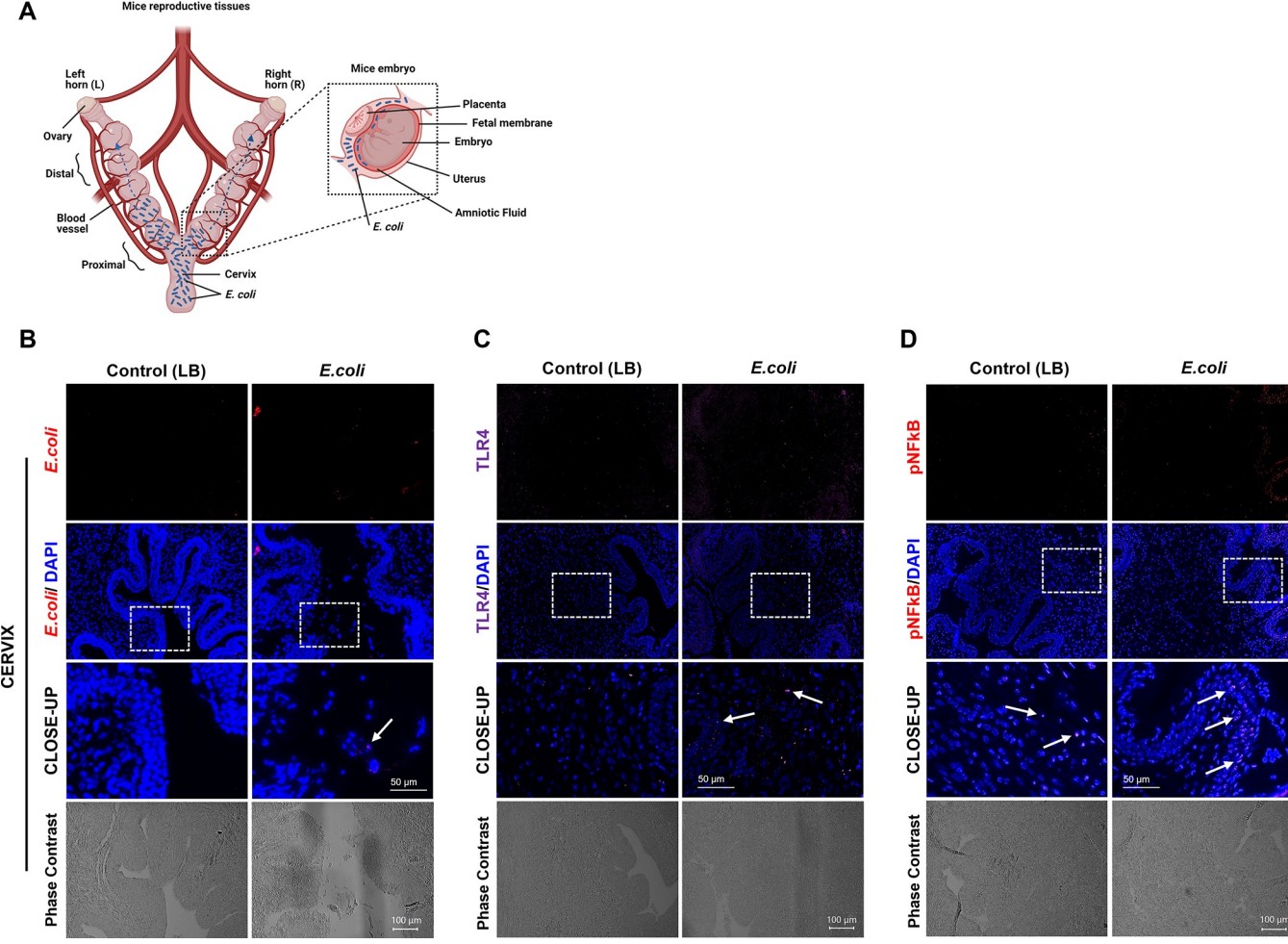

**Fig 4. *E. coli* induced inflammation in mouse reproductive tissues.** (**A**) Schematic illustration of mouse reproductive tissues and close-up display of an embryo in the uterine cavity. Reprinted from biorender under a CC BY license, with permission from biorender, original copyright. (**B-D,** N = 3). Immunohistochemical analysis of cervix collected 6 hr after vaginal administration of $10^{11}$ CFU of *E. coli*. *E. coli* detected in the cervical sections *(**B**, see white arrow, E. coli positive staining)*. Infected cervix shows higher expression of TLR-4 (**C**) and P-NFkB (**D**) compared to control cervix (Scale bar, 100 μm). The close-up displays the enlarged tissue area marked by white boxes (Scale bar, 50 μm).

## *E. coli* induces histologic chorioamnionitis (HCA)

Next, we examined induction of HCA, a classic inflammatory signature in the fetal membranes, indicative of severity of infection and host inflammatory response by the presence of neutrophils (ly-6G+ cells). For determining HCA, Ly-6G staining was performed on membranes collected at 6 and 24 hr. As shown in **Fig 6**, a few Ly-6G positive cells were seen in tissues from liquid broth injected animals, indicative of normally resident neutrophils in the membranes [51, 65]. We have shown that normal fetal membranes (in humans) have ~7% CD45+ cells of which neutrophils are a predominant constituent [65]. A slight increase, but not significant numbers, in Ly-6G+ cells were seen at 6 hr in the fetal membranes of animals injected with *E. coli* (**Fig 6B**) that was increased substantially within 24 hr (**Fig 6A and 6B**).

## Discussion

PTB and pPROM are associated with MIAC and IAI [3, 4, 6, 25]. However, these conditions are often clinically diagnosed very late and management strategies focus primarily to delay

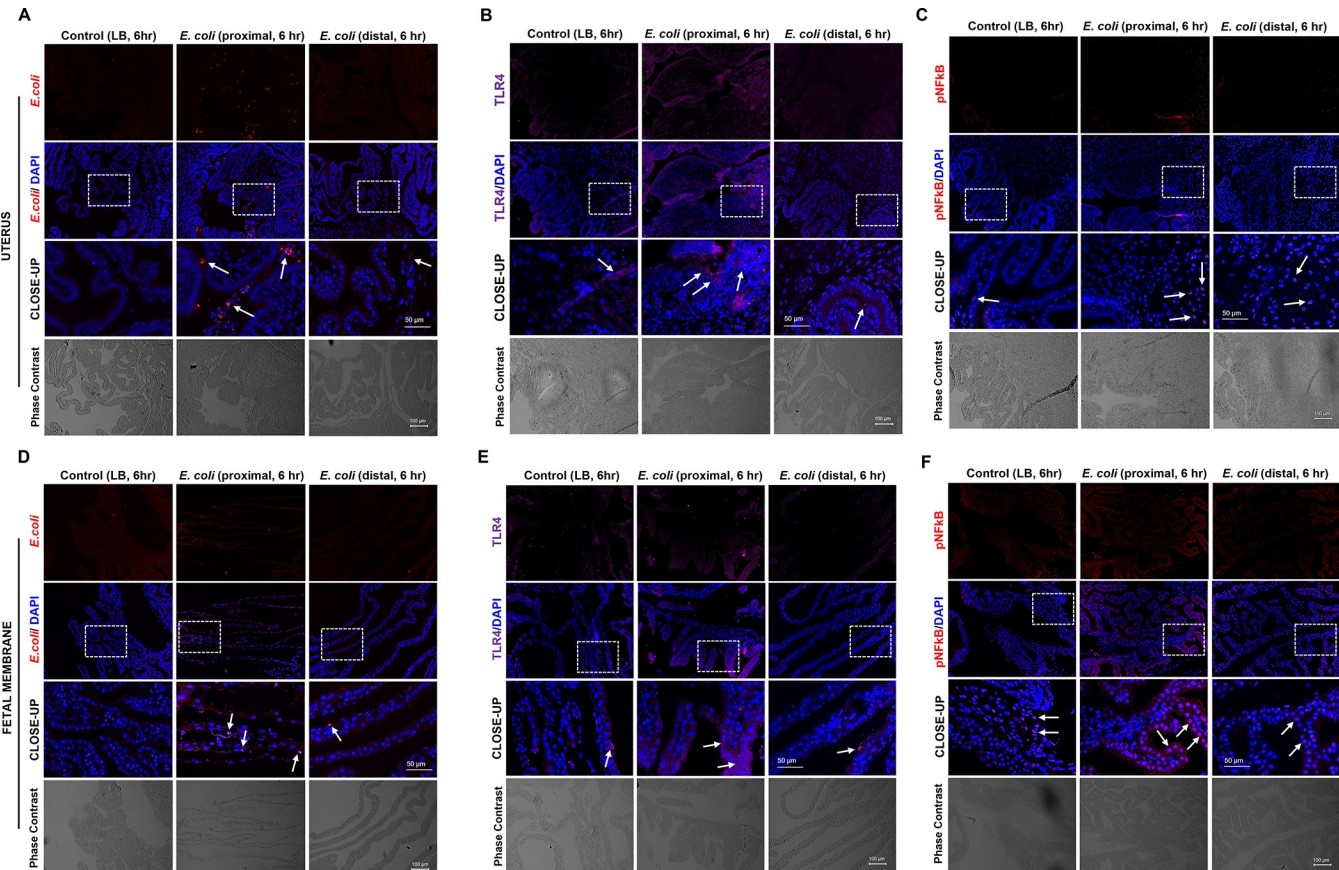

**Fig 5. *E. coli* induced inflammatory marker stepwise progression is consistent with ascending bacterial infection in mice uterine cavity.**
Immunohistochemical analysis proximal and distal uterine (**A-C**) and fetal membrane tissues (**D-F**) collected 6 hr after vaginal administration of $10^{11}$ CFU of *E. coli* (N = 3). Proximal portions of the uterine sections show higher rates of *E. coli* than distal portions (**A**). Inflammatory markers, TLR-4 and P-NF-κB expressions were higher in the proximal uterine section than distal sections (**B-C**). Control uterine sections show comparable expression of the inflammatory markers (TLR-4 and P-NF-κB) to distal sections (**B-C**). Proximal fetal membrane section shows higher rate of *E. coli* than distal portion (**D**). Inflammatory markers, TLR-4 and P-NFkB expressions were higher in the proximal fetal membrane section than distal sections (**E-F**), and control sections show comparable expression of the inflammatory markers (TLR-4 and P-NFkB) to distal sections (**E-F**). (Scale bar, 100 μm). The close-up displays the enlarged tissue area marked by white boxes (Scale bar, 50 μm).

labor by a few hours or days for administration of antibiotics or steroids. Unfortunately, these approaches have not reduced the risk of prematurity (low birth weight < 2,500 grams) or morbidities associated with PTB [1, 66, 67]. To improve pregnancy outcomes, better models that can provide mechanistic evidence of establishment of infection and development of inflammation causing PTB and or pPROM are needed. Using an ascending model of infection of *E. coli*, we determined the following: 1. We developed an ascending infection model where vaginal inoculation of *E. coli* produced a dose dependent pregnancy outcome (PTB). Higher dose of *E. coli* ($10^{10}$ CFU) caused PTB in 48 hr compared to lower doses ($10^3$ CFU and $10^6$ CFU) that delivered near term, 2. After high dose infection, *E coli* was localized in cervix, uterus and in the amniotic cavity of pups proximal to cervix within 6 hr and in amniotic fluid and tissues from distal horns within 24 hr. 3. Ascending high dose infection induced TLR-4 activation, a ligand for LPS (Gram negative [*E. coli*] cell wall component) and activated proinflammatory transcription factor P-NF-κB expression. Both TLR-4 and P-NF-κB were co-localized in tissues along with *E. coli*, 4. Activation of inflammatory markers was not widespread in 6 hr; however, all feto-maternal uterine tissues showed signs of inflammation within 24 hr, 5.

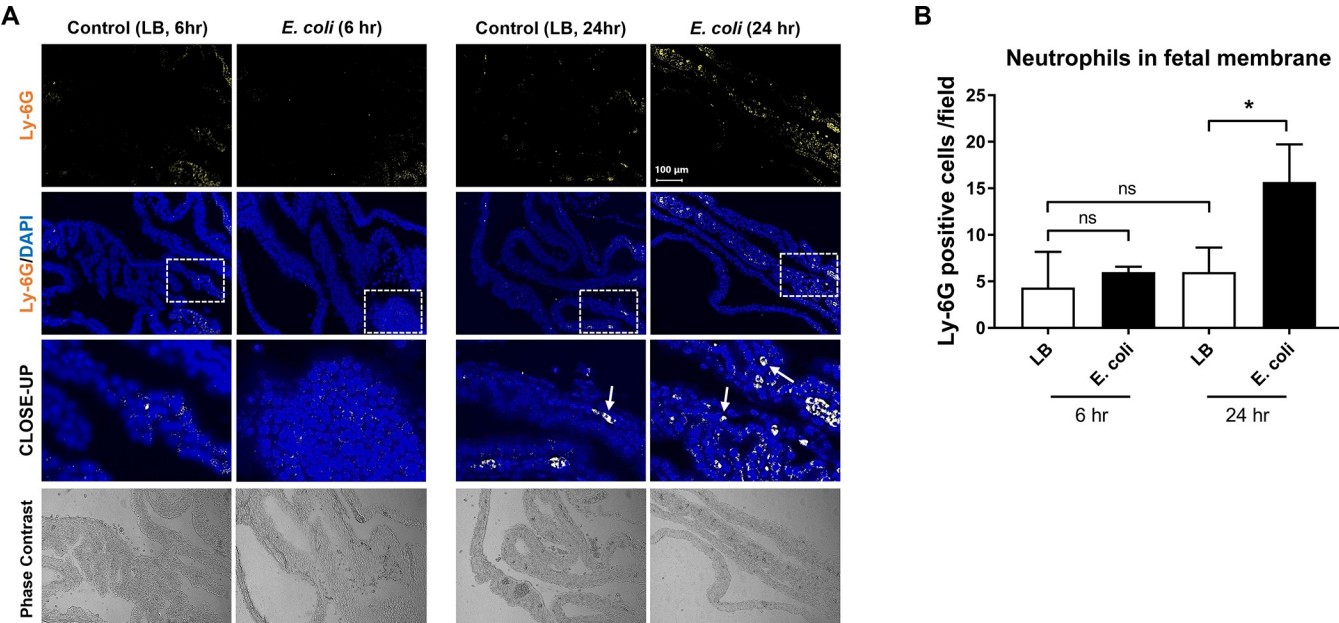

**Fig 6. *E. coli* induced neutrophil infiltration in the fetal membranes.** A. Immunohistochemistry analysis of mice fetal membranes collected from 6 and 24 hr after $10^{11}$ CFU of *E. coli* vaginal administration (N = 3). Neutrophil infiltration detected at 24 hr after the bacterial infection in the fetal membrane tissues (white arrows pointing to Ly-6G positive cells). However, 6 hr after bacterial infection, neutrophil detection was comparable to non-infected controls. (Scale bar, 100 μm). B. Quantification of neutrophils in fetal membrane in (A). Number of Ly-6G positive cells per random fields (N = 3). The data are presented as means ± SEM.* *P* = 0.0219, paired t-test.

Inflammation of all horns and feto-maternal tissues preceded preterm delivery and 6. MIAC and IAI induced histologic chorioamnionitis where fetal membranes of infected animals showed higher number of neutrophil infiltrations compared to control animals. This model not only replicated the results from Suff *et al.* [56] but determined the kinetics of microbial ascension further. We were able to show the tissue specific inflammation and development HCA with our model. Fetal tissue (lung, brain, heart etc.) testing for infection and/or inflammation associated changes was not attempted in this study. We conclude that the model described here showed natural progression of infection and development of inflammation leading to PTB. This model is suitable to study MIAC and IAI associated mechanistic pathways in PTB.

To introduce experimental rigor and to test validity of our data, a subset of our animals was injected with gentamicin (20 mg/kg) 4 and 24 hr after the *E. coli* injection. Gentamicin after 4 hr led to 100% term delivery whereas gentamicin after 24 hr delayed PTB in 40% of animals (**S6 Fig**). Multiple inferences can be made from this outcome: 1. Although invasion of amniotic cavity may begin as early as 6 hr of injection in this model, establishment infection in all fetal sacs does not occur until 24 hr. Administration of antimicrobial agents prior to establishment of IAI may reduce the risk of PTB, 2. Similarly, inflammation is also limited at early stages; however, HCA and increased inflammatory marker expressions seen at 24 hr diminished antimicrobials' effect to mitigate PTB. The condition observed at 24 hours is often faced in high-risk clinics and late administration of antimicrobials neither delays nor minimizes tissue inflammation including HCA.

Although our model established MIAC and IAI associated PTB, multiple challenges still remain to further define and mitigate the process in humans. 1. The exact timing of a pregnant subject getting infected is difficult to assess unless there is a clinical indicator 2. The kinetics of

MIAC in humans (subclinical infection) is difficult to predict, especially during a polymicrobial infection, and 3. Inflammation and inflammatory markers show tremendous heterogeneity and therefore, a biomarker indicative of an underlying specific infectious etiology is difficult to assess. In several instances, non-infectious (sterile infections) etiologies also show similar inflammatory biomarker profiles both in humans and mouse models [61, 68–71].

Current biomarker clinical trials in humans have not generated a serum marker specifically indicative of early signs of preterm birth [72–75]. Therefore, novel strategies of biomarker research have been discussed and several potential approaches have been discussed to predict high risk pregnancies at very early stages pregnancy [76] and the studies of animal models, such as the one we present here, could potentially guide future human studies that will be critical in delineation of clinical signs and predictive markers specific for very early infection. Some of these strategies include, cervico-vaginal microbiome, cell free mRNA and proteome-based biomarkers in maternal plasma, fetal membrane and placental cells in maternal circulation, and fetal and maternal exosomes and their cargo profile are a few such approaches [76]. Although these biomarkers are in their early stages of discovery and validation, many of these markers have yielded promising results to show that they may predict high risk pregnancies as early as first trimester [77–80]. Future studies of animal models, such as the one we present here, may be critical in delineation of clinical signs and predictive markers specific for very early infection.

There are several animal models reported for PTB; however, only limited number of articles had experimental models and approaches that can yield data to understand microbial invasion and potential mechanisms [59, 81, 82]. Although no animal models completely mimic human parturition, preclinical mouse models have provided valuable information regarding mechanisms as seen in humans to design future human trials. This model also has limitations as we did not perform live imaging of microbial ascension and *E. coli* is not the most common microbial pathogen associated with PTB and pPROM. Therefore, this model needs to be further tested with microbes that are more commonly associated with MIAC, IAI and PTB (e.g., genital mycoplasmas, Gardnerella). Recently, we developed an organ-on-a-chip (OOC) model of ascending infection [83]. Using multiple cells from the feto-maternal interface, we were able to demonstrate the kinetics of ascending infectious stimulus and generation of inflammatory mediators in response to a stimulus [83]. OOC model is developed to overcome certain limitations associated with animal models, 2D cell cultures, transwell models and organ explant models and it maintains intercellular interactions, flow of biochemical between tissues and generate scenarios like that seen in utero. These models need further development and validation and simultaneous testing with animal models to show that human cell based OOCs can adequately replace animal models to study pregnancy complications. This new step, along with models such as delineated here, may enhance translatability to the human condition.

In summary, the animal model presented here is reproducible and provides a model for testing various ascending infection of various severities. Clinically, these models can generate a knowledge base from which to understand mechanisms of infection associated preterm birth, determine targets for intervention or identify potential biomarkers that can predict a high-risk pregnancy status early in during pregnancy.

## Supporting information

**S1 Fig. *E. coli* induced preterm birth (PTB) in a dose dependent manner.** Higher doses of *E. coli* ($10^{11}$ CFU and $10^{10}$ CFU) delivered significantly shorter time frame compared to control (LB) ($P<0.001$ and $P = 0.002$, respectively). Low dose of *E. coli* ($10^6$ CFU and $10^3$ CFU) delivered in shorter time frame compared to control, however not significantly ($P = 0.17$ and

$P$ = 0.3, respectively). CFU-colony forming unit.
(TIF)

**S2 Fig. CFSE-stained *E. coli* ascending infection in fetal membrane and uterine tissues collected from 6 hr, 24 hr, and 48 hr after $10^{10}$ CFU dose (N = 3).** Scale bar, 50 μm.
(TIF)

**S3 Fig. Higher rate *E. coli* detected in pups from proximal part than distal part 6 hr after $10^{11}$ CFU of *E. coli* vaginal administration (N = 3).** Scale bar, 200 μm.
(TIF)

**S4 Fig. *E. coli* growth on MacConkey agar.** Representative images of amniotic fluid recovered bacterial growth on MacConkey agar. (A) Control plate. Negative for *E. coli* growth. (B) Positive *E. coli* growth plate.
(TIF)

**S5 Fig. Bacterial culture from maternal blood collected from control (LB) and $10^{11}$ CFU of *E. coli* administered mice. (A)** MacConkey's agar culture showed no bacterial growth (N = 3). (**B**) Confirmational culture in the liquid nutritional broth using the samples directly transferred from MacConkey's agar culture from **A**.
(TIF)

**S6 Fig. Gentamicin treatment reduced preterm birth rate.** Gentamicin (20 mg/kg) at 4 hr after *E. coli* administration showed 100% prolonged gestation to term delivery compared to controls (PBS only) ($P<0.001$) (N = 3); however, same dose given after 24 hours shows 40% effect on increased length of gestation to term delivery compared to controls ($P<0.001$) (N = 5).
(TIF)

## Author Contributions

**Conceptualization:** Enkhtuya Radnaa, Ananth Kumar Kammala, Samantha Sheller-Miller, Ramkumar Menon.

**Data curation:** Enkhtuya Radnaa, Tuvshintugs Baljinnyam, Ananth Kumar Kammala.

**Formal analysis:** Nicholas R. Spencer, Enkhtuya Radnaa, Tuvshintugs Baljinnyam, Ananth Kumar Kammala.

**Funding acquisition:** Ramkumar Menon.

**Investigation:** Nicholas R. Spencer, Enkhtuya Radnaa, Talar Kechichian, Ourlad Alzeus G. Tantengco, Ananth Kumar Kammala, Samantha Sheller-Miller.

**Methodology:** Enkhtuya Radnaa, Tuvshintugs Baljinnyam.

**Project administration:** Enkhtuya Radnaa, Ananth Kumar Kammala, Ramkumar Menon.

**Resources:** Tuvshintugs Baljinnyam, Elizabeth Bonney, Ramkumar Menon.

**Supervision:** Enkhtuya Radnaa, Ramkumar Menon.

**Validation:** Nicholas R. Spencer, Tuvshintugs Baljinnyam.

**Visualization:** Nicholas R. Spencer, Enkhtuya Radnaa.

**Writing – original draft:** Nicholas R. Spencer, Enkhtuya Radnaa, Tuvshintugs Baljinnyam, Ramkumar Menon.

**Writing – review & editing:** Nicholas R. Spencer, Enkhtuya Radnaa, Tuvshintugs Baljinnyam, Ramkumar Menon.

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
