## [Decision Letter · Decision Letter 0]

5 Aug 2021

PONE-D-21-16450

Development of a mouse model of ascending infection and preterm birth

PLOS ONE

Dear Dr. Radnaa,

Thank you for submitting your manuscript to PLOS ONE. After careful consideration, we feel that it has merit but does not fully meet PLOS ONE’s publication criteria as it currently stands. Therefore, we invite you to submit a revised version of the manuscript that addresses the points raised during the review process.

We look forward to receiving your revised manuscript.

Kind regards,

Jefferson Terry

Academic Editor

PLOS ONE

Journal Requirements:

2. To comply with PLOS ONE submissions requirements, in your Methods section, please provide additional information on the animal research and ensure you have included details on methods of analgesia, efforts to alleviate suffering and general animal welfare.

4. We note that Figure 3A and 4A in your submission contain copyrighted images. All PLOS content is published under the Creative Commons Attribution License (CC BY 4.0), which means that the manuscript, images, and Supporting Information files will be freely available online, and any third party is permitted to access, download, copy, distribute, and use these materials in any way, even commercially, with proper attribution. For more information, see our copyright guidelines: http://journals.plos.org/plosone/s/licenses-and-copyright.

a. You may seek permission from the original copyright holder of Figure 3A and 4A to publish the content specifically under the CC BY 4.0 license. 

Additional Editor Comments:

Reviewers' comments:

Reviewer's Responses to Questions

**Comments to the Author**

1. Is the manuscript technically sound, and do the data support the conclusions?

Reviewer #1: Yes

Reviewer #2: Yes

2. Has the statistical analysis been performed appropriately and rigorously? 

Reviewer #1: Yes

Reviewer #2: No

3. Have the authors made all data underlying the findings in their manuscript fully available?

Reviewer #1: Yes

Reviewer #2: Yes

4. Is the manuscript presented in an intelligible fashion and written in standard English?

Reviewer #1: Yes

Reviewer #2: Yes

5. Review Comments to the Author

Reviewer #1: Spencer et al.’s manuscript describes an interesting E.coli ascending infection model of preterm birth. The following comments will help to improve the manuscript.

Methods

There is lots of variation amongst infection-related mouse models of preterm birth. In this study, mice were inoculated at E15, what was the rationale behind choosing this gestation time point?

Results

Figure 1 – I would change the scale of the X axis on this graph to show the gestation at delivery more clearly. I prefer Sup Fig1 to show this.

Figure 2- the clarity of the images is poor, are you able to improve this?

Figure 3 – the piecharts in 3B are confusing, I think it would be more clear if this data were shown as % bar charts with 6hr and 24hrs adjacent to each other. Also blue= sterile is not strictly true, so “E.coli growth”/“no growth” is probably better.

Line 274 – where is the fetus infected, it is not very clear from your figure? Could there be contamination or is bacteria seen within the fetus?

Line 310 – change farthest horns to farthest pups in the distal horn

Figure 4 – it is not clear from your legend and your figure what reproductive tissues have been stained, can you please clarify. What exactly does close-up mean, please can you label with microscope magnification.

Figure 4, Figure 5 & 6 should be combined and the labels/legends should be more clear as to what organ has been stained. The pictures are also very blurry, would it be possible to improve the quality of them?

Line 325 & 328 – this sentence is confusing, have you done a formal cell count? If not, then this sentence is misleading.

Line 361 – neutrophil counts would be an important additional experiment here and should be included to improve the paper.

Discussion

Line 390 This statement is not strictly true = Suff et al,’s model shows inflammatory cytokine expression in the uteroplacental tissues and Suff et al., 2020 PMID: 32117260 describes microbial ascension within the E.coli model.

Line 404 – where is this data?

Line 406 – please provide a reference for this statement as this is a bold statement that I am not sure is necessarily true clinically? I don’t believe there is evidence that antimicrobials do not delay inflammation/HCA in women presenting with PTB, particularly in the PPROM cohort (see PMID: 32591087).

Line 416 – preterm birth is missing here, I also think this statement is not necessarily true as quantitative fetal fibronectin is being used clinically in some parts of the world with good prediction PMID: 25932845

Line 415 – I agree that these models are important to understand mechanisms and test therapies but I am not convinced that they will provide biomarkers as murine anatomy/immunology etc is so different, large human studies are key for determining significant biomarkers for PTB.

Reviewer #2: This is a nice, compact, methods paper that outlines the specific timeline and anatomic progression of ascending infection following vaginal inoculation with EColi. The studies are well designed and I have no issue with any of the methods. Given that many animal models of preterm birth use this model or a similar model- it is useful to have well delineated data regarding the time line and confirming actual microbiologic invasion. This will be a useful underpinning to grant applications, study design etc.

A brief rationale of why the specific inflammatory and neutrophil markers were selected would be helpful. Statistically, given small numbers and lack of demonstrated normal distribution - non parametric methods are preferred. The introduction is three pages long - it should probably be cut by about 1/3. The length of the discussion is appropriate. Figure 4a may not be needed is space is an issue - Figure 2 does a nice job of showing the uterine horns. For Figure 2 - it would be helpful to explain DiD in legend.

6. PLOS authors have the option to publish the peer review history of their article (what does this mean?). If published, this will include your full peer review and any attached files.

Reviewer #1: No

Reviewer #2: No

---

## [Author Response · Author response to Decision Letter 0]

14 Sep 2021

Reviewer #1: Spencer et al.’s manuscript describes an interesting E.coli ascending infection model of preterm birth. The following comments will help to improve the manuscript.

Methods

There is lots of variation amongst infection-related mouse models of preterm birth. In this study, mice were inoculated at E15, what was the rationale behind choosing this gestation time point?

E15 mice were chosen for the following reasons: 1) this time point primarily mimic late second trimester-early third trimester period in humans, a highly vulnerable period during pregnancy for development of adverse events, 2) Placental development is not complete in this model until E10 and adequate time period is provided for pregnancy to fully develop and mature 3) it has been reported that senescence, EMT and inflammatory cytokines peak around E18, and senescence-associated inflammation starts in fetal membrane and maternal tissues [PMID: 27324096; PMID: 31814178]. Therefore, a period before these developments are chosen to represent a homeostatic period of intrauterine tissues where pregnancy is considered normal and an induction of labor by infectious agents can be monitored for an adequate period of time prior to the beginning of normal labor process. and 4) Moreover, our reported analysis of inflammatory changes indicated that inflammation related characteristics starts to develop after E15. In this model maternal cervix remodeling and uterine activation was observed by E17 which is the indication of the maternal reproductive tract preparation for the parturition at term and therefore a period before this will represent a balanced inflammatory state that will allow us to determine infection induced changes [PMID: 30679631] [PMID: 30679631]. 

Results

Figure 1 – I would change the scale of the X axis on this graph to show the gestation at delivery more clearly. I prefer Sup Fig1 to show this.

Thank you for recommending making Fig.1 clearer. We have changed the X-axis of Fig.1 to make it similar to Sup. Fig.1. 

Figure 2- the clarity of the images is poor, are you able to improve this?

Thank you for the comment. We have improved the clarity of Fig.2. 

Figure 3 – the piecharts in 3B are confusing, I think it would be more clear if this data were shown as % bar charts with 6hr and 24hrs adjacent to each other. Also blue= sterile is not strictly true, so “E.coli growth”/“no growth” is probably better.

Thank you for your comment. We have changed green=infected to “E.coli growth”; blue=sterile to “no growth” as suggested. Pie charts were originally chosen because they parallel the sequential clockwise plating used for the amniotic fluid cultures. We were also asked to represent similar data as pie charts by the Editors of various journals. Therefore, we have not changed. We are more than happy to change if the editor requests.

Line 274 – where is the fetus infected, it is not very clear from your figure? Could there be contamination or is bacteria seen within the fetus?

Thank you for the comment. The resolution of the IVIS was approach will not distinguish bacterial cells and hence we use tagged microbial cells to track their ascent. Therefore, microbial particle level clarity is hard to obtain. To overcome this limitations we have used two additional approaches to determine microbial invasion 1) We performed microbial culture using amniotic fluid samples from proximal fetal compartment. Here we showed E. coli growth in amniotic fluid and 2) Immunohistochemical staining for E coli showed positive staining in the pup (Fig. 3B and Sup. Fig.3, respectively). 

Line 310 – change farthest horns to farthest pups in the distal horn 

We have now changed as suggested. 

Figure 4 – it is not clear from your legend and your figure what reproductive tissues have been stained, can you please clarify. What exactly does close-up mean, please can you label with microscope magnification. 

We have placed tissue name on the Fig.4 to make it clear. Close-up means the enlarged image from the marked white boxes. We have inserted scale bars on the close-up images and clarified what close-up is in the figure legend as well. 

Figure 4, Figure 5 & 6 should be combined and the labels/legends should be more clear as to what organ has been stained. The pictures are also very blurry, would it be possible to improve the quality of them? 

We have combined the Fig. 5 & Fig. 6 as recommended. We have inserted scale bars on the close-up images and clarified what close-up is in the figure legend as well. Close-up images are to make images look clearly. 

Line 325 & 328 – this sentence is confusing, have you done a formal cell count? If not, then this sentence is misleading. 

We have edited the sentence to read “Positive staining for inflammatory markers were detected in sections from E. coli-injected animals than in sections from control media-injected animals. 

Line 361 – neutrophil counts would be an important additional experiment here and should be included to improve the paper.

We have quantified the Ly-6G positive cells in the fetal membrane tissues, and incorporated the data to Fig. 6 (edited Fig.6; old Fig.7) as suggested. 

Discussion

Line 390 This statement is not strictly true = Suff et al,’s model shows inflammatory cytokine expression in the uteroplacental tissues and Suff et al., 2020 PMID: 32117260 describes microbial ascension within the E.coli model. 

We have edited the Line 390 to: This model not only replicated the results from Suff et al.56 but determined the kinetics of microbial ascension further. We were able to show the tissue specific inflammation and development HCA with our model. 

Line 404 – where is this data?

We have removed the sentence from the discussion since we did not show the data. 

Line 406 – please provide a reference for this statement as this is a bold statement that I am not sure is necessarily true clinically? I don’t believe there is evidence that antimicrobials do not delay inflammation/HCA in women presenting with PTB, particularly in the PPROM cohort (see PMID: 32591087).

 We have rephrased our sentence to: “The condition observed at 24 hours is often faced in high-risk clinics and late administration of antimicrobials neither delays nor minimizes tissue inflammation including HCA.” Since mouse models do not replicate pPROM conditions, it may not be appropriate to compare these data with a pPROM cohort of Dr Kacerovsky.

 Multiple systematic reviews and meta-analysis of randomized control trials have shown that prophylactic antibiotics do not reduce the risk of PTL, pPROM, and PTB [ PMID: 17877593; PMID: 19956761; PMID: 24307518; PMID: 25621770]. The American College of Obstetricians and Gynecologists also recommends that women in preterm labor should not be treated with antibiotics for the sole purpose of preventing preterm delivery [PMID: 12738177]. 

Line 416 – preterm birth is missing here, I also think this statement is not necessarily true as quantitative fetal fibronectin is being used clinically in some parts of the world with good prediction PMID: 25932845

 We have put the PTB which was missing. 

We have rephrased the sentence to: “Current biomarker clinical trials in humans have not generated a serum marker specifically indicative of early signs of preterm birth”. 

We agree that successful prediction has been made Andy Shennan’s group in UK with quantitative Ffn and a few others. It has not yet received widespread acceptance in many clinical practices. Fetal fibronectin has been shown to have good negative predictive value, however its positive predictive value for preterm birth is currently lacking when used alone. [ACOG practice bulletin no. 127 PMID: 22617615]

Line 415 – I agree that these models are important to understand mechanisms and test therapies but I am not convinced that they will provide biomarkers as murine anatomy/immunology etc is so different, large human studies are key for determining significant biomarkers for PTB.

 Thank you for the comment. We have rephrased the sentence to: “Therefore, novel strategies of biomarker research have been discussed and several potential approaches have been discussed to predict high risk pregnancies at very early stages pregnancy76 , and the studies of animal models, such as the one we present here, could potentially guide future human studies that will be critical in delineation of clinical signs and predictive markers specific for very early infection.” 

Reviewer #2: This is a nice, compact, methods paper that outlines the specific timeline and anatomic progression of ascending infection following vaginal inoculation with EColi. The studies are well designed and I have no issue with any of the methods. Given that many animal models of preterm birth use this model or a similar model- it is useful to have well delineated data regarding the time line and confirming actual microbiologic invasion. This will be a useful underpinning to grant applications, study design etc.

A brief rationale of why the specific inflammatory and neutrophil markers were selected would be helpful. Statistically, given small numbers and lack of demonstrated normal distribution - non parametric methods are preferred. The introduction is three pages long - it should probably be cut by about 1/3. The length of the discussion is appropriate. Figure 4a may not be needed is space is an issue - Figure 2 does a nice job of showing the uterine horns. For Figure 2 - it would be helpful to explain DiD in legend.

 Thank you for your nice comments. Since the readership of PLoS ONE is beyond the general OB audience, we felt that it is necessary to provide adequate background prior to introducing the topic. Therefore, we have not reduced the length.

We have previously shown that after LPS treatment, Ly-6G positive neutrophil cells in mice fetal membrane were detected higher compared to those of the PBS control group [DOI: 10.1126/sciadv.abd3865]. 

We have provided the DiD explanation in the Fig.2 legend.

---

## [Decision Letter · Decision Letter 1]

9 Nov 2021

Development of a mouse model of ascending infection and preterm birth

PONE-D-21-16450R1

Dear Dr. Radnaa,

We’re pleased to inform you that your manuscript has been judged scientifically suitable for publication and will be formally accepted for publication once it meets all outstanding technical requirements.

Jefferson Terry

Academic Editor

PLOS ONE

Reviewers' comments:

Reviewer's Responses to Questions

**Comments to the Author**

1. If the authors have adequately addressed your comments raised in a previous round of review and you feel that this manuscript is now acceptable for publication, you may indicate that here to bypass the “Comments to the Author” section, enter your conflict of interest statement in the “Confidential to Editor” section, and submit your "Accept" recommendation.

Reviewer #1: All comments have been addressed

2. Is the manuscript technically sound, and do the data support the conclusions?

Reviewer #1: Yes

3. Has the statistical analysis been performed appropriately and rigorously? 

Reviewer #1: Yes

4. Have the authors made all data underlying the findings in their manuscript fully available?

Reviewer #1: Yes

5. Is the manuscript presented in an intelligible fashion and written in standard English?

Reviewer #1: Yes

6. Review Comments to the Author

Reviewer #1: I am happy with the author's response to my comments - I think this manuscript should be published.

7. PLOS authors have the option to publish the peer review history of their article (what does this mean?). If published, this will include your full peer review and any attached files.

Reviewer #1: No

---

## [Editor Report · Acceptance letter]

19 Nov 2021

PONE-D-21-16450R1 

Development of a mouse model of ascending infection and preterm birth 

Dear Dr. Radnaa:

I'm pleased to inform you that your manuscript has been deemed suitable for publication in PLOS ONE. Congratulations! Your manuscript is now with our production department. 

Kind regards, 

on behalf of

Dr. Jefferson Terry 

Academic Editor

PLOS ONE